# Modeling SARS-CoV-2 propagation using rat coronavirus-associated shedding and transmission

**Caroline J. Zeiss**[1]*, **Jennifer L. Asher**[1], **Brent Vander Wyk**[2], **Heather G. Allore**[2,3], **Susan R. Compton**[1]

1 Department of Comparative Medicine, Yale School of Medicine, New Haven, CT, United States of America,
2 Department of Internal Medicine, Yale School of Medicine, New Haven, CT, United States of America,
3 Department of Biostatistics, Yale School of Public Health, New Haven, CT, United States of America

* caroline.zeiss@yale.edu

**Data Availability Statement:** All relevant data are within the manuscript and its Supporting Information files.

## Abstract

At present, global immunity to SARS-CoV-2 resides within a heterogeneous combination of susceptible, naturally infected and vaccinated individuals. The extent to which viral shedding and transmission occurs on re-exposure to SARS-CoV-2 is an important determinant of the rate at which COVID-19 achieves endemic stability. We used Sialodacryoadenitis Virus (SDAV) in rats to model the extent to which immune protection afforded by prior natural infection via high risk (inoculation; direct contact) or low risk (fomite) exposure, or by vaccination, influenced viral shedding and transmission on re-exposure. On initial infection, we confirmed that amount, duration and consistency of viral shedding, and seroconversion rates were correlated with exposure risk. Animals were reinfected after 3.7–5.5 months using the same exposure paradigm. 59% of seropositive animals shed virus, although at lower amounts. Previously exposed seropositive reinfected animals were able to transmit virus to 25% of naive recipient rats after 24-hour exposure by direct contact. Rats vaccinated intranasally with a related virus (Parker's Rat Coronavirus) were able to transmit SDAV to only 4.7% of naive animals after a 7-day direct contact exposure, despite comparable viral shedding. Cycle threshold values associated with transmission in both groups ranged from 29–36 cycles. Observed shedding was not a prerequisite for transmission. Results indicate that low-level shedding in both naturally infected and vaccinated seropositive animals can propagate infection in susceptible individuals. Extrapolated to COVID-19, our results suggest that continued propagation of SARS-CoV-2 by seropositive previously infected or vaccinated individuals is possible.

## Introduction

As the COVID-19 pandemic proceeds, SARS-CoV-2 must navigate an increasingly heterogeneous immune landscape. Individual immunity to SARS-CoV-2 infection may be gained through natural infection or vaccination. The former route, otherwise known as infection-

**Funding:** All authors were supported by the National Science Foundation under the RAPID mechanism (NSF 2031950; 20-052; Zeiss, PI). HGA and BV are partially supported by the National Institute on Aging at the National Institutes of Health (P30AG021342). Content is solely the responsibility of the authors and does not necessarily represent official views of the National Science Foundation or National Institutes of Health.

**Competing interests:** The authors have declared that no competing interests exist.

induced herd immunity, is widely regarded as an ineffective strategy [1]. This route achieves unpredictable immunity [2–5] and incurs substantial morbidity [6] and mortality [5, 7]. Consequently, mass vaccination against SARS-CoV-2 is underway as the safest and most effective means of controlling the COVID-19 pandemic [8]. For either route, our understanding of the extent to which previously naturally exposed [9–11] or vaccinated [12] individuals can shed and transmit virus on re-exposure is presently emerging. Immunologic heterogeneity following natural infection [10, 11, 13], variable efficacy of different vaccines [8], as yet unclear duration of immunity [14–17] and emergence of new variants [18] are critical determinants of the level of herd immunity needed to control COVID-19. Because of these variables, herd immunity needed to eradicate COVID-19 is likely to be difficult to achieve [3, 8, 19].

Predicting the path of SARS-CoV-2 to endemic status can be aided by study of other human [20–22] and animal [23] coronaviruses. Sialodacryoadenitis virus (SDAV) is a highly infectious betacoronavirus [24, 25] that infects the upper respiratory tract [26], lacrimal and salivary glands [27, 28] and lung [28–30] of rats. SDAV infection results in a two-week course of asymptomatic to mild respiratory disease [29, 30]. Consequently, it is best suited to model transmission dynamics of COVID-19 [31], rather than disease pathogenesis. Like SARS CoV-2 [31–35], SDAV infection can be transmitted by asymptomatically infected individuals [23] via airborne, direct contact or fomite routes [24]. Both SARS-CoV-2 and SDAV can persist on hard surfaces for up to 28 days [34, 36] and 2 days [37] respectively. SDAV resides in the subgenus Embecovirus, and is most closely related to mouse hepatitis virus (MHV) and two human upper respiratory pathogens, human coronavirus HKU1 (HCoV-HKU1) and Human coronavirus OC43 (HCoV-OC43) [38–40]. HCoV-OC43 and HCoV-HKU1 cause annual wintertime outbreaks of mild respiratory illness [21], and their transmission characteristics have been used to accurately predict population spread and seasonal recurrence of SARS-CoV-2 [22].

We used a similar approach to develop a rat model of SARS-CoV-2 transmission using SDAV with the goal of understanding the extent to which individuals with natural or vaccine-induced immunity can shed and transmit virus on re-exposure. The longevity of immunity imparted by vaccination or natural infection with SARS-CoV-2 is not yet clear [17, 41, 42]. Based on the premise that coronaviruses may share common characteristics regarding immunity, researchers have examined reinfection rates of seasonal coronaviruses to gain insight into this issue [20]. Like HCoV-OC43 and HCoV-HKU1 in humans [20], immunity in rats elicited by SDAV is temporary [43–45], thus allowing us to model this variable during reinfection. Beginning with a defined SDAV inoculum, we modeled heterogeneous viral exposure in a naturally infected population using a range of high (inoculation and direct contact) and low (fomite) risk exposures. Recovered animals were then re-exposed to SDAV to determine the role of naturally acquired immunity in subsequent viral shedding and transmission to naïve animals. Data from naturally infected re-exposed animals were compared to that from animals exposed to SDAV after infection with Parker's Rat Coronavirus (RCV), a closely related coronavirus [46, 47]. RCV has been shown to elicit protective cross-immunity to SDAV [43], and is used in this context as a heterologous vaccine for SDAV.

## Materials and methods

### Virus amplification and quantification

SDAV (strain 681) was isolated at Yale in 1976. RCV (strain 8190) was originally obtained from the American Type Culture Collection, Rockville, MD [43]. Stocks of SDAV and RCV were generated in L2p176 cells [48, 49]. Briefly, confluent L2.p176 cells were pretreated for 1 hour with 75ug/ml of DEAE-D in 75% DMEM 25% L15 media. Cells were incubated with

virus diluted in 75% DMEM 25% L15 with trypsin for 1 hour and cell/media was harvested at 3 days post-infection. Viral titers were determined by plaque assay [48]. Briefly, 6 well plates of L2p176 cells were pretreated with 75ug/ml DEAE-D in 75% DMEM/25% L15 media for 3 hours. Cells were rinsed with PBS and inoculated with 10- fold dilutions of virus in 75% DMEM/25% L15 with 75ug/ml DEAE-D and trypsin. One hour later, inocula was removed, cells were rinsed with PBS and were overlaid with 0.55% Seaplaque agarose/minimal media/trypsin. Three days post-inoculation, cells were fixed with formalin, agarose was removed and plaques were visualized with Giemsa.

## Animals and housing

Seven week-old female and male SAS outbred Sprague-Dawley rats (150-250g) were purchased from Charles River Laboratories (Wilmington, MA). Animals were housed (separated by sex) in Tecniplast (West Chester, PA) individually ventilated cages (GR900 for rats) that provide high microbial biocontainment. Sentinel animals were placed on each side of the rack and tested every 3 months for antibodies to rodent pathogens, including SDAV, with consistently seronegative results. Rooms were maintained at 72˚F on an evenly split light cycle (7AM:7PM). Animals were housed on corncob bedding, had access to autoclaved pellets (2018S, Envigo, Somerset, NJ) and acidified water ad lib, and were acclimated for 5–7 days prior to infection. Animals were individually identified using ear tags. These were regularly inspected and replaced as needed. Viral inoculation and animal handling was performed in a Class II biosafety cabinet. All exposure groups were separated by sex. All animal work was conducted under an approved Yale Animal Use and Care Committee protocol. Yale University is accredited by the Association for Assessment and Accreditation of Laboratory Animal Care

## Anesthesia, viral inoculation and euthanasia

Rats were anesthetized briefly using the open drop method (isoflurane: propylene glycol 30% v/v). Intranasal inoculation of $2X10^e4$ plaque-forming units (pfu) SDAV or $1X10^e3$ pfu RCV in Dulbecco's Modified Eagle Medium (DMEM) was performed in a total volume of 50µl per animal (25 µl per nostril). Animals were fully recovered within 2–3 minutes of inoculation. At the end of reinfection and transmission experiments, animals were euthanized using 70% carbon dioxide.

## Initial infection with SDAV (Fig 1A)

Four exposure groups were defined for initial SDAV infection:

a. <u>Inoculated rats (n = 19, 47% female).</u> Animals were inoculated intranasally with $2X10^e4$ pfu SDAV in 50µl DMEM, followed by individual housing in a clean cage for 48 hours.

b. <u>Direct contact (n = 31, 48% female):</u> Naïve animals (1–3 animals) were placed with one inoculated rat in a new clean cage 48 hours post inoculation. After 24 hours, exposed rats were separated from the inoculated rat placed in a new clean cage.

c. <u>Fomite contact animals (n = 55; 53% female).</u> Naïve animals (2–3 animals) were placed in a dirty cage (containing contaminated bedding, furniture, food and water nipple) that had been inhabited by an inoculated rat for 48 days post inoculation. After 24 hours, exposed rats were relocated to a new clean cage in small groups of 2–3 animals (fomite cohabitation group; n = 30; 53% female) or singly (fomite single group; n = 25; 52% female).

d. <u>Mock (n = 10):</u> A control group (n = 10; 50% female) was inoculated with DMEM alone.

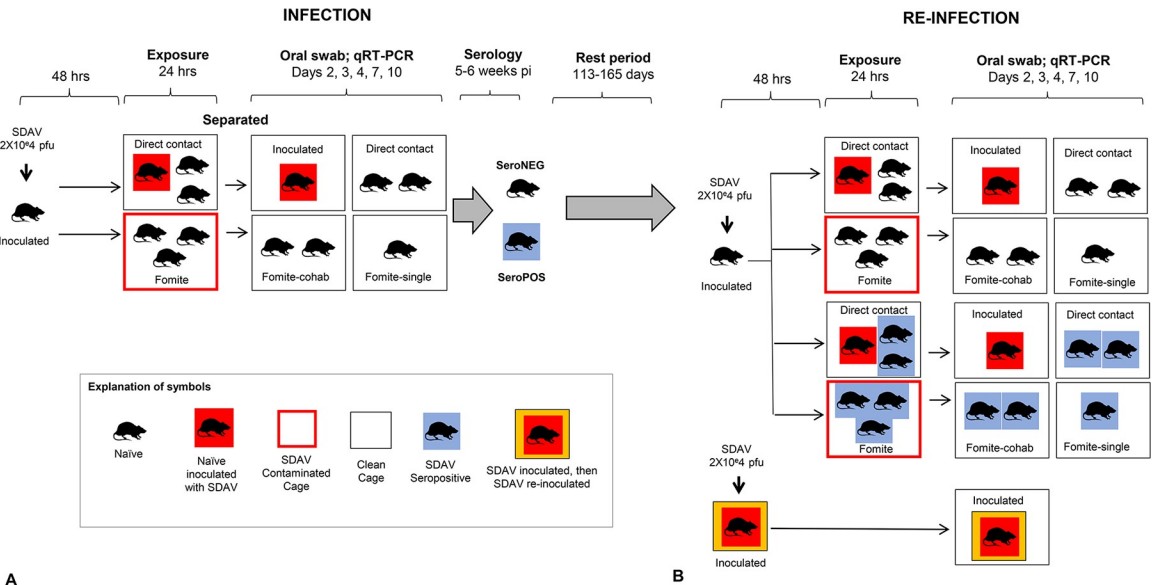

**Fig 1. SDAV infection and re-infection paradigm. A. Initial infection with SDAV.** Naïve animals were inoculated intranasally with 2X10ᵉ4 pfu SDAV. After 48 hours, inoculated animals or their dirty cages were used to expose naïve animals to the virus. For direct contact exposure, one inoculated animal was placed in a clean cage with naïve animals for 24 hours. For fomite exposure, naïve animals were placed in a dirty cage that had been inhabited by an inoculated rat for 24 hours. After 24 hours, all rats were relocated to new clean cages to constitute four groups: inoculated rats, direct exposure rats, and two fomite exposure groups–a fomite cohabitation group constituting 2–3 animals, and a fomite single group with only one animal. Oral swabs were taken on Day 2, 3, 4, 7 and 10 days post-exposure (dpe) on all animals, and serology performed 5–6 weeks later. A control group (not shown) was inoculated with DMEM alone, and similarly swabbed and bled. After initial exposure, animals assumed either seronegative or seropositive status. All groups were evenly split by sex. **B. Reinfection with SDAV.** Naïve seronegative rats were inoculated intranasally with 2X10ᵉ4 pfu SDAV to provide a source of infection. Rats that had originally received intranasal infection with SDAV were re-infected intranasally again with the same viral dose. Seropositive and seronegative animals from the initial infection experiment were randomly assigned direct contact, fomite contact-cohabitation, and fomite contact-singly housed contact groups for their second exposure. Time between initial and second exposure ranged from 113–165 days. Oral swabs were taken on Day 2, 3, 4, 7 and 10 dpe on all animals. Animals were sacrificed at 10 dpe and assessed for seroconversion. All groups were evenly split by sex.

Oral swabs were taken on day 2, 3, 4, 7 and 10 days post-exposure (dpe: defined as inoculation or exposure via a contact mode) on all animals (thus inoculated rats began this process 96 hours before other exposure groups). Each animal was weighed on its day of viral exposure (Day 0) and on every day on which oral swabs were taken. Gloves and equipment were sterilized with 200ppm MB-10 between each animal. Health checks were performed daily for 10 days post-exposure.

### Reinfection with SDAV (Fig 1B)

A total of 106 animals were used in the reinfection study, of which 40 were seronegative and 66 were seropositive (Table 2). Rats that had originally received intranasal infection with SDAV (inoculated rats) were re-infected intranasally again with the same viral dose (n = 18). Naïve seronegative rats (mock inoculated rats from the prior experiment or naïve purchased rats) received intranasal inoculations to provide a source of infection for remaining animals (n = 18). Remaining animals (22 seronegative rats and 48 seropositive rats) were randomly assigned direct contact, fomite contact-cohabitation, and fomite contact-singly housed groups for their second exposure. Exposure and testing paradigms were identical to those described for initial reinfection. Time between initial and second exposure ranged from 113–165 days. Animals were evenly split by sex and aged 6–7 months at sacrifice.

## Assessing transmission of SDAV to naïve rats by previously SDAV infected rats (Fig 2A)

Inoculated rats that had received an initial dose of $2X10^e4$ pfu SDAV, and had subsequently shed virus and seroconverted, received a second similar intranasal inoculation 113–165 days later (n = 13). These animals were placed in a clean cage with susceptible recipient rats 48 hours post inoculation (direct contact paradigm). Recipient rats were age and sex-matched seronegative rats from the initial SDAV infection experiment that had never tested SDAV positive on oral swabs following fomite exposure (n = 13). After 24 hours, exposed recipient rats were separated from the inoculated rat and placed in a new clean cage. Body weights and oral swabs were taken on both groups of animals at 2, 3, 4, 7 and 10 dpe, followed by serologic testing of naïve animals. Animals were evenly split by sex and aged 6–7 months at sacrifice.

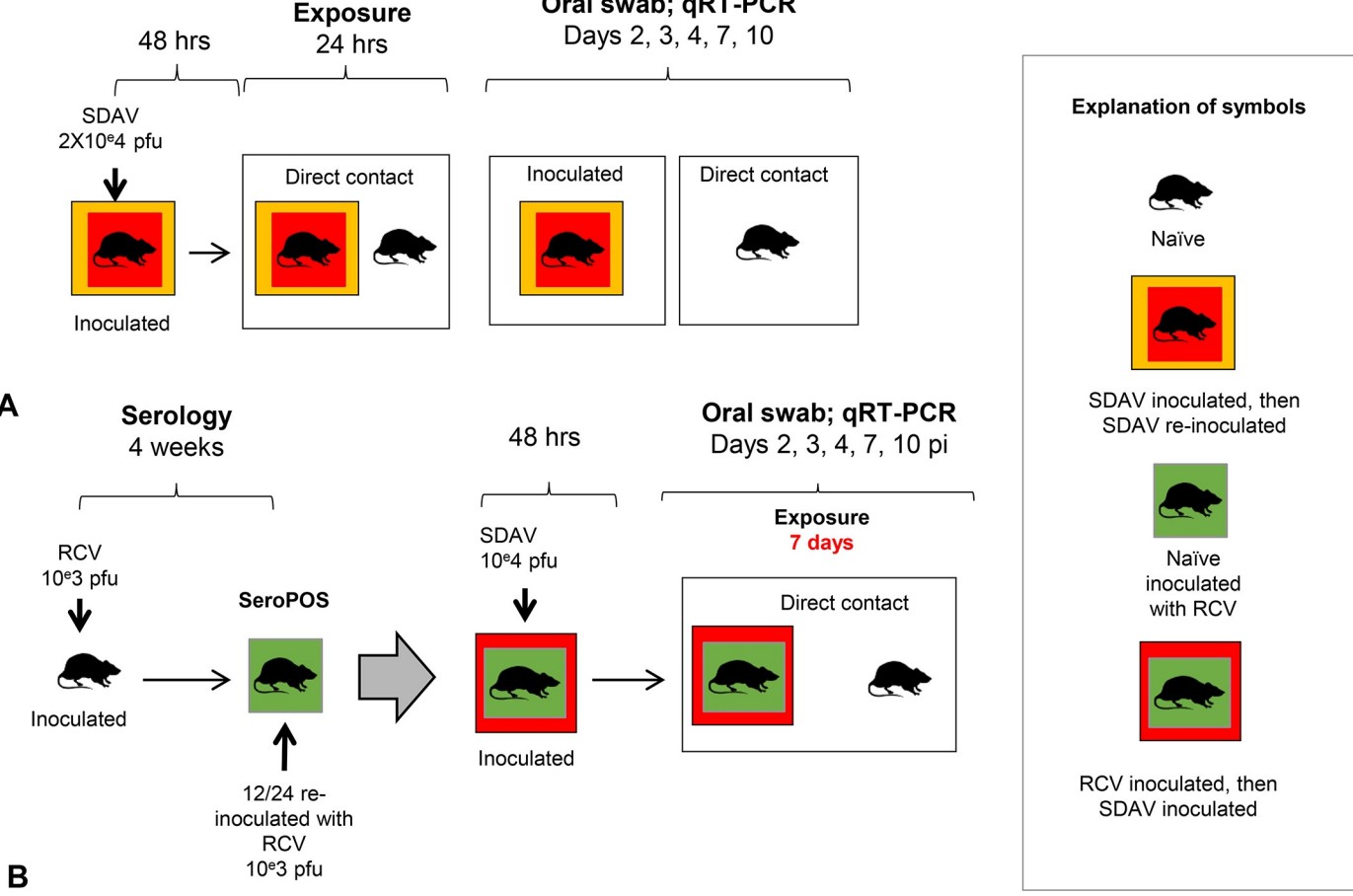

**Fig 2. Transmission of SDAV to naïve rats by SDAV-exposed or RCV-vaccinated rats. A. Transmission of SDAV to naïve rats by previously SDAV infected rats.** Inoculated rats that had received an initial dose of $2X10^e4$ pfu SDAV, and had subsequently shed virus and seroconverted, received a second similar intranasal inoculation 112–140 days later (n = 13). After 48 hours, these animals were placed in a clean cage with susceptible recipient rats (n = 13; direct contact paradigm). After 24 hours, recipient rats were separated from the inoculated rat and placed in a new clean cage. Body weights and oral swabs were taken on both groups of animals at 2, 3, 4, 7 and 10 dpi. Animals were sacrificed at 10 dpe and assessed for seroconversion. All groups were evenly split by sex. **B. Transmission of SDAV to naïve rats by RCV vaccinated rats.** Eight-week-old naïve rats (n = 24) were inoculated intranasally with $1X10^e3$ pfu RCV and assessed for seroconversion 4 weeks later. All animals seroconverted, and were divided into a single vaccine group (n = 12) and a double vaccine group (n = 12). This latter group received an additional RCV inoculation at the same dose. Six weeks after their last RCV exposure, seropositive animals were inoculated with $2X10^e4$ pfu SDAV. After 48 hours, these animals were co-housed in a clean cage with a susceptible rat (n = 12 for single vaccine group; n = 9 for double vaccine group) for 7 days. Body weights and oral swabs were taken on both groups of animals at 2, 3, 4, 7 and 10 dpe. Animals were sacrificed at 10 dpe and assessed for seroconversion. All groups were evenly split by sex.

## Modeling vaccination using RCV (Fig 2B)

Parker's rat coronavirus (RCV) is a spontaneously occurring rat virus that is closely antigenically related to SDAV [46]. Heterologous vaccination with RCV confers significant but not absolute cross-protection against subsequent challenge with SDAV [43]. First, we performed a dose-finding study to assess the dose of RCV that would impart protection to subsequent challenge with SDAV. Three groups of male rats (n = 3/group) were inoculated with $10^e3$, $10^e4$ or $10^e5$ pfu RCV in 50 μl of DMEM. Seroconversion was confirmed in all 2 weeks after inoculation. Two weeks later, all animals were challenged with intranasal $2X10^e4$ pfu SDAV (50 μl). Low viral shedding (Cq 30.5–30.7 cycles for one day only over a 10-day period) was noted in only two animals at the higher RCV inoculation groups ($10^e4$ or $10^e5$ pfu RCV). Based on these data, a dose of $10^e3$ RCV pfu RCV in 50 μl of DMEM was selected for the vaccination study. Eight-week-old naïve purchased male and female SD rats were inoculated intranasally either once (n = 12) or twice with a month interval between inoculations (n = 12). Seroconversion was confirmed one month after the first inoculation. Rats vaccinated with one or two doses of RCV were challenged by intranasal inoculation with $2X10^e4$ pfu SDAV in 50 μl of DMEM six weeks after their last RCV exposure. After 48 hours, one SDAV-inoculated rat was co-housed with one naïve rat for 7 days (n = 12 for single vaccine group; n = 9 for double vaccine group). Body weights and oral swabs were taken on both groups of animals at 2, 3, 4, 7 and 10 dpe, followed by serologic testing of naïve animals at 10 dpe. Animals were evenly allocated by sex and aged 4–4.5 months at sacrifice.

## Assessing oral shedding of SDAV by semi-quantitative RT-PCR

Rats were swabbed orally using sterile flocked swabs (Hydraflock, Puritan Medical Products, Guilford, ME) to confirm viral shedding by quantitative RT-PCR. 600 ul of RLT buffer (Qiagen) was added to each swab and the sample was vortexed. 350 ul of 70% ethanol was added to 350 ul of lysed sample in RLT and the mixture was transferred to the RNeasy mini column. RNA was extracted following the manufacturer's instructions and RNA was eluted from the column with 50 ul of RNase-free water. 2.5 ul of RNA was amplified using the iTaq Universal SYBR Green One Step kit (Biorad) and the following primers (SD29629:AGAAAACGCCGGTAGCA GAA and SD30197:CCTTCCCGAGCCTTCAACAT) using a Biorad CFX Connect Real-time System. Primers were designed in house. Numbers correspond to the nucleotide positions in the SDAV genome (JF92616.1). The reaction conditions were 10 min at 50C; 5 min at 95C; and 40 cycles of 10 sec at 95C, 20 sec at 59.1C and 36 sec at 72C. All assays contained negative and positive controls. PCR positivity was defined as a rat having a Cq of < 40 for at least 1 observation

*Serology*: 10 days to six weeks after exposure to SDAV or RCV, animals were bled to assess seroconversion. Sera was tested for coronaviral antibodies using an indirect immunofluorescence assay [50]. Briefly 20 ul of sera diluted tenfold in PBS was placed on a glass slide containing fixed L2p176 cells infected with mouse hepatitis virus strain S. Bound rat antibodies were detected with FITC-conjugated goat anti-rat IgG (Jackson ImmunoResearch).

## Data analysis and statistics

Descriptive statistics were conducted using t-tests, non-parametric tests of medians, and chi-squares tests for proportions where appropriate with data analyzed using regression models. Continuous outcomes (e.g Cq or body weight) used standard linear models with an autoregressive covariance parameter to control for the repeated measurements within animal while count outcomes used Poisson regression with a log-link. Raw data are included as S1–S6 Figs.

## Results

### Initial infection with SDAV

**Body weight.** Compared to mock-inoculated animals, SDAV-inoculated rats experienced declines in weight gain at days 2–4 post infection (**S1 Fig**), with no differences by sex. Only inoculated rats gained significantly less quickly than mock inoculated control animals (p < .001). Apart from transient porphyrin staining of eyes lasting less than 24 hours in 5 rats, no other clinical signs were noted.

**Viral shedding.** All SDAV-inoculated rats and those exposed via direct contact tested PCR positive on oral swabs (**Table 1**), with declining proportions of PCR-positivity in fomite–exposed animals that were co-housed (73.3%) or singly housed (24%), suggesting that subsequent rat-rat transmission occurred in co-housed animals. Route of exposure significantly influenced amount of viral shedding (p<0.0001; **S2 Fig**). Compared to shedding in inoculated and direct contact groups, viral shedding following fomite exposure was significantly lower (p< .0001)

Route of exposure significantly influenced duration of viral shedding (defined as the number of observations, in days, where shedding was present; p<0.0001; **S3 Fig**). Inoculated and direct exposure groups shed virus for approximately twice as long as fomite exposed animals (significant at p<0.0001 for fomite-cohabitation group only). Fomite exposed animals shed intermittently; however, shedding more commonly persisted to 10 dpe (**S4 Fig**). Consistency of shedding was significantly affected by exposure mode (p < .0001), with inoculated and direct exposure groups shedding with significantly greater consistency than fomite-cohabitation (p < .0001) and fomite single groups (p < .0001).

**Seroconversion.** All SDAV-inoculated and direct contact rats seroconverted. Seroconversion declined in cohabiting and singly housed fomite-exposed animals (50% and 8% respectively). These two exposure groups also experienced discordant results between PCR testing and seroconversion (**Table 1**). These data indicate that viral shedding detectable by PCR does not invariably result in seroconversion. Conversely, because viral shedding is intermittent, a negative PCR test does not rule out the potential of infection sufficient to induce an immune response. However, seroconversion was significantly associated with greater amounts (p<0.0001) and duration (p<0.0001) of viral shedding. (**S5 Fig**). Sex did not significantly affect viral shedding amount, duration or seroconversion.

**Table 1. Viral shedding and seroconversion following initial infection with SDAV.**

| Exposure type | Duration of exposure | PCR positive | Cq (mean, range) | Shedding time points (mean, range) | Sero+ | Sero- | PCR Neg, Sero+ | PCR Pos, Sero- |
|---|---|---|---|---|---|---|---|---|
| SDAV 2X10e4 pfu (n = 19)* | Inoculation | 19/19 (100%) | 31.5; 25.2–37.4 | 3.1; 2–4 | 19 (100%) | 0 | 0 | 0 |
| Direct contact (n = 31) | 24 hours | 31/31 (100%) | 31.3; 25.5–36.0 | 2.9; 1–3 | 31 (100%) | 0 | 0 | 0 |
| Fomite-cohab (n = 30) | 24 hours | 22/30 (73.3%) | 32.9; 27.4–36.7 | 1.5; 1–3 | 15 (50%) | 15 (50%) | 9 (30%) | 9 (30%) |
| Fomite-single (n = 25) | 24 hours | 6/25 (24%) | 33.2; 28.6–36.2 | 1.5;1–4 | 2 (8%) | 22 (88%)** | 1 (5%) | 4 (16%) |
| Media inoculation (n = 10) | Inoculation | 0/10 (0%) | Neg, all >40 | 0 | 0 | 10 (100%) | 0 | 0 |

Cq = quantification cycle. Only animals with viral shedding (Cq<40 cycles) are included in Cq and shedding time calculations. Observation time points comprised post-exposure days 2, 3, 4, 7, 10.

*One rat in the inoculation group died of unrelated causes after blood was taken for serology.

** One rat in the fomite single group died of unrelated causes before blood was taken for serology.

**Re-infection with SDAV.** Rats were aged for 113–165 days before reinfection with SDAV by inoculation, direct contact or fomite exposure. Amount of viral shedding (expressed as lowest observed Cq) was significantly influenced by both exposure mode (p < .01) and serologic status (p < .001; S6 Fig). Similarly, duration of shedding was significantly influenced by both exposure mode (p < .05) and serologic status (p < .001; S6 Fig). A lower proportion of previously SDAV-exposed seropositive rats (38.9% compared to 100%) shed less virus (p < .001) for a shorter period (p < .001) compared to previously mock-inoculated seronegative rats when re-inoculated with the same dose of SDAV. This clearly demonstrates protective immunity (that did not however eliminate viral shedding) on reinfection with the same virus and same dose.

To investigate the differential role of exposure mode on amount of viral shedding, rats were stratified by serological status (Table 2). Among seronegative rats, as with initial infection, exposure mode significantly affected amount (p < .001) and duration (p < .0001) of viral shedding. Compared to seronegative SDAV-inoculated rats, direct contact rats shed less virus (p = .06) for a shorter period of time (p<0.001). Fomite cohabitation animals shed less virus (p < .01) for a shorter period of time (p<0.001).

In seropositive animals, immunity obtained from prior SDAV exposure via a variety of routes altered this pattern. Direct contact and fomite cohabiting rats experienced higher, but not significant (p = .09) levels of shedding than the SDAV-inoculated seropositive rats previously exposed to SDAV via the same route. In contrast, exposure mode was significantly (p < .01) associated with duration of shedding. SDAV-inoculated rats generally shed for fewer

**Table 2. Viral shedding after SDAV reinfection in seropositive and seronegative groups.**

**A. Seropositive (n = 66)**

| Re-exposure type | Duration between infections | Duration of exposure | PCR pos | Cq (mean, range) | Shedding time points (mean, range) | Sero+ | Sero- | PCR Neg, Sero+ | PCR Pos, Sero- |
|---|---|---|---|---|---|---|---|---|---|
| SDAV 2X10e4 pfu intranasal (n = 18)* | 113–144 days | Inoculation | 7/18 (38.9%) | 32.7; 28.9–36.4 | 1.7;1–4 | - | - | - | - |
| Direct contact (n = 19) | 143–165 days | 24 hours | 15/19 (78.9%) | 34.1; 29.1–36.6 | 2; 1–4 | - | - | - | - |
| Fomite-cohab (n = 17) | 114–143 days | 24 hours | 13/17 (76.5%) | 34.4; 30.7–37.9 | 1.7; 1–2 | - | - | - | - |
| Fomite-single (n = 12) | 114–165 days | 24 hours | 4/12 (33.3%) | 34.5; 33.0–36.2 | 1.7; 1 | - | - | - | - |

**A. Seronegative (n = 40)**

| | | | | | | | | | |
|---|---|---|---|---|---|---|---|---|---|
| SDAV 2X10e4 pfu intranasal (n = 18)** | 82–163 days | Inoculation | 18/18 (100%) | 30.6; 23.5–35.3 | 3; 2–4 | 18 (100%) | 0 | 0 | 0 |
| Direct contact (n = 9) | 143–165 days | 24 hours | 8/9 (88.8%) | 31.9; 27.3–33.4 | 1.4; 1–3 | 8 (88.8%) | 1 (11.1%) | 1 (11.1%) | 1 (11.1%) |
| Fomite-cohab (n = 9) | 165 days | 24 hours | 6/9 (66.6%) | 31.3; 27.2–34.5 | 1.6;1–4 | 9 (100%) | 0 | 0 | 0 |
| Fomite-single (n = 4) | 165 days | 24 hours | 1/4 (25%) | 33.9; 33.9–34.0 | 2; 2 | 1 (25%) | 3 (75%) | 1 (25%) | 3 (75%) |

*Seropositive group: Animals given SDAV 2X10e4 pfu on initial infection were reinfected via the same route. All other animals were randomized between initial and subsequent routes of infection and were exposed to naïve animals given SDAV 2X10e4 pfu via direct contact or fomite exposure.

**Seronegative group: Animals given SDAV 2X10e4 pfu reinfection were previously mock infected controls or naïve purchased animals. All other animals were exposed to naïve animals inoculated with SDAV 2X10e4 pfu via direct contact or fomite exposure.

- Seropositive animals were not re-tested for seroconversion after reinfection

Cq = quantification cycle. Only animals with viral shedding (Cq<40 cycles) are included in Cq and shedding time calculations. Observation time points comprised post-exposure days 2, 3, 4, 7, 10.

**Table 3. Transmission of SDAV by previously SDAV infected or RCV vaccinated animals.**

**A. Transmission after SDAV reinfection**

| Source animals (n = 13) | | | | | Susceptible recipient animals (n = 13) | | | | | |
|---|---|---|---|---|---|---|---|---|---|---|
| *Initial infection* | *Re-infection* | *Duration between infection* | *PCR pos* | *Cq; shed time points (mean; range)* | *Duration of exposure* | *PCR pos* | *Cq; time points shedding* | *Sero+* | *PCR Neg, Sero+* | *PCR Pos, Sero-* |
| SDAV 2X10e4 pfu | SDAV 2X10e4 pfu | 113–140 days | 2/13 (15.4%) | 32.2 (29.2–34.4)<br><br>1.5 (1–2) | 24 hours | 3/13 (23.1%) | 34.0 (32.7–36.5)<br><br>2.3 (1–4) | 3 (23.1%) | 1 (7.7%) | 1 (7.7%) |

**B. Transmission after RCV single vaccination**

| Source animals (n = 12) | | | | | Susceptible recipient animals (n = 12) | | | | | |
|---|---|---|---|---|---|---|---|---|---|---|
| RCV 10e3 pfu | SDAV 2X10e4 pfu | 48 days | 4/12 (33.3%) | 32.5(30.1–34.5)<br>1.8(1–3) | 7 days | 1/12 (8.3%) | 31.8(28.5–25.5)<br>3(0) | 1 (8.3%) | 0 | 0 |

**C. Transmission after RCV double vaccination**

| Source animals (n = 12) | | | | | Susceptible recipient animals (n = 9) | | | | | |
|---|---|---|---|---|---|---|---|---|---|---|
| RCV 10e3 pfu twice, 30 days apart | SDAV 2X10e4 pfu | 42 days | 7 (58.3%) | 34.7(29.5–37.6)<br>1.6(1–3) | 7 days | 2 (22.2%) | 32.5(32.3–32.9)<br>1.5(1–2) | 0 (0%) | 0 | 0 |

Viral shedding and seroconversion in naïve (susceptible recipient) animals exposed to SDAV inoculated animals.

Cq = quantification cycle. Only animals with viral shedding (Cq<40 cycles) are included in Cq and shedding time calculations. Observation time points comprised post-exposure days 2, 3, 4, 7, 10.

observations, and significantly fewer observations than the direct contact group (p < .05). These data indicate that immunity obtained via direct contact or fomite exposure was largely protective regardless of how it was obtained, but that some heterogeneity in protection against duration of shedding was imparted by route of initial and subsequent exposure.

**Viral transmission by SDAV reinfected or RCV vaccinated rats (Table 3).** Transmission was defined in two ways 1) if the target rats exhibited shedding on any observation and 2) if the target rat seroconverted. Following SDAV inoculation, no significant differences in number of animals shedding virus, amount of virus shed or duration of shedding were noted across vaccine groups, indicating that one or two doses provided equivalent protection. Transmission rates from the SDAV reinfected (n = 13) and RCV vaccinated rats (n = 21, collapsing 1 and 2 doses) were compared to each other and to a reference group consisting of the previously described rats in direct contact with SDAV naïve rats (n = 31) using chi-square tests. Regardless if shedding or seroconversion was treated as the metric of transmission, proportions in the both SDAV reinfected source group (3/13, shedding and seroconversion) and RCV vaccinated group (3/21, shedding; 1/21, seroconversion) were significantly lower than the SDAV naïve direct contact reference group (31/31, shedding and seroconversion;p < .001) in both cases. However, the proportions did not differ significantly between SDAV reinfected and RCV vaccinated rats (shedding p = .65; seroconversion, p = .27).

## Discussion

It appears increasingly likely that SARS-CoV-2 will persist as an endemic virus shaped by immune dynamics that regulate reinfection [51, 52]. The extent to which humans with natural or vaccine-induced immunity, who are re-exposed to SARS-CoV-2, can shed and transmit virus is only just emerging [12, 53]. Absent these data, researchers have utilized established patterns of seasonal coronavirus transmission to model patterns of SARS-CoV-2 persistence during the post-pandemic period [20–22]. We have employed the same approach by using

SDAV, a rat respiratory rat coronavirus phylogenetically and biologically closely related to HCoV-OC43 and HCoV-HKU1, to model coronaviral transmission in animals. Of particular interest to us was the extent to which immune protection afforded by prior high (inoculation and direct contact) to low (fomite) risk exposure or vaccination influenced the amount of virus shed on re-exposure. To study this, we employed a natural exposure setting to track transmission dynamics initiated by known starting inoculum, thus providing controlled *in vivo* data suitable for use in future SEIRS (susceptible, exposed, infected, recovered, susceptible) modeling [54]. A key issue confronting control of the COVID-19 pandemic is whether low amounts of shed virus detectable by common screening tests such as PCR can achieve transmission [55]. Therefore, we also assessed whether low amounts of shed virus in re-exposed animals could infect susceptible animals. We used the two most widely used measures of SARS-CoV-2 (and other viral) surveillance, reverse-transcriptase polymerase chain reaction (RT-PCR) testing [56] and serology [57] to examine the relationships between exposure routes, seroconversion and viral shedding. Shedding in rats was assessed using oral swabs. Salivary gland infection has been demonstrated for SARS-CoV-2 [58], with saliva-based testing gaining traction as a gold-standard diagnostic sample [59].

Amount and duration of viral shedding was significantly influenced by route of exposure, and was highest in inoculated and direct contact groups, and lowest in fomite groups. Fomite transmission was amplified by subsequent cohabitation of rats, suggesting that transmission risk of this route can be amplified by close contact such as dense co-housing conditions. As in the human population, the viral dose that resulted in infection of an individual cannot be directly measured in natural exposure settings. However, data from SARS-CoV-2 studies indicate that higher infectious doses of virus are implicated in higher risk of transmission [60]. Infectious dose is assumed to be much higher in direct contact compared to fomite settings, corresponding to respective high and low transmission risk of these exposure types [61, 62]. Similarly, higher viral loads in COVID-19 patients are associated with more reliable seroconversion [63]. Our results are consistent with these data and imply that higher viral exposure results in a greater amount and duration of viral shedding, followed by more consistent seroconversion. Conversely, low viral load accompanying fomite exposure may result in a positive viral PCR test but fail to elicit an antibody response.

Following initial infection with SDAV, 59% of seropositive animals shed virus on re-exposure after 3.7–5.5 months, although at lower levels than in initial infection. Shedding rates on reinfection in our model are much higher than those reported for human SARS-CoV-2 [9]. This may reflect a true biological difference between SDAV and SARS-CoV-2. However, our entire population was re-exposed followed by timed repeated PCR testing, thus maximizing the likelihood of detecting shedding. Re-exposure events in a human population are rarely known, thus precluding coordination of testing with the exposure event. Therefore, the actual shedding on re-exposure in human populations may be higher than reported.

It remains unclear to what extent detection of viral genetic material by PCR in mucosal swabs translates directly to transmissibility [64–66]. With SARS-CoV-2, live viral culture positivity declines with increasing cycle threshold values [66], and consequently, live viral shedding can be inferred from Cq values on PCR testing [67]. Cycle threshold values of 33–34 cycles reflect low enough live viral shedding to render patients non-contagious [32, 65, 66]. Lower levels of viral shedding and even lower levels of live virus detection in animals challenged with SARS-CoV-2 after vaccination have been noted in minks [68], hamsters [64] and macaques [64]. The extent of this reduction, particularly regarding nasal shedding, is determined by dose and route of vaccine administration in animals [64, 69]. In humans, viral shedding after vaccination at relatively high levels (Cq values <25 cycles) is associated with the SARS-CoV-2 Delta variant [53].

To further understand the relationship between PCR-detectable viral shedding and potential for transmission, we assessed the risk of transmission by animals shedding low amounts of virus following SDAV re-exposure or following heterologous vaccination. Consistent with prior rat studies [44, 45], rats reinfected with SDAV via the same route (intranasal inoculation given 113–140 days apart) were able to induce viral shedding and/or seroconversion in 25% of susceptible contact rats after 24-hour exposure by direct contact. Observed cycle threshold (Cq) values in reinfected source rats ranged from 29–34 cycles with short shedding durations (1–2 days). It should be noted that in all three instances where susceptible recipient rats seroconverted, viral shedding by SDAV re-infected rats was not detected, implying that shedding sufficient for transmission can occur for short periods that may escape detection. While both SARS-CoV-2 [58] and SDAV infect salivary glands and can be detected in the oral cavity, we recognize that oral swabs in our rats may not have detected shedding via other routes e.g. nasal shedding.

Next, we assessed transmission by animals exposed to SDAV shortly after heterologous vaccination with Parker's Rat Coronavirus (RCV), a betacoronavirus that is closely related to SDAV [46]. In prior studies, infection with RCV results in cross-protective seroconversion, and disease protection following subsequent SDAV infection [43]. While a significant proportion of vaccinated animals (11/24) shed SDAV at low amounts (range 29.5–36 cycles), they achieved transmission in only one recipient after 7 days of direct contact exposure. These results indicate that protection against transmission after reinfection in the immediate post vaccination period is superior to that several months after natural infection. We did not test whether this protection would decline over months at the same rate as that afforded by SDAV inoculation, however some decline is expected from previous studies [43]. Data from both groups taken together imply that Cq values above 29 are associated with transmission.

## Conclusion

As the COVID-19 pandemic proceeds, the virus must navigate an increasingly heterogeneous immune landscape of naïve, naturally infected and vaccinated individuals. Predicting its path to endemic status can be aided by study of other endemic human coronaviruses such as HCoV-OC43 and HCoV-HKU1. A common element allowing useful comparisons across coronaviruses is their tendency to cross the species barrier and follow a pandemic-to-endemic trajectory characterized by temporary immunity and declining disease severity [20, 40, 70–72]. To generate controlled transmission data in an animal model, we utilized SDAV, a rat Embecovirus which is closely related to HCoV-OC43 and displays this typical epizootic to enzootic transmission pattern [23, 44].

Shedding and seroconversion following initial natural SDAV infection was heterogeneous and influenced by route of exposure. Viral shedding on re-exposure was much lower than on initial exposure, but was nevertheless able to result in transmission to susceptible individuals. Vaccination imparted greater protection against transmission after SDAV challenge, however this protection would be expected to decline over a time period similar to that imparted by infection with SDAV. If these data are extrapolated to SARS-CoV-2 transmission, it appears that viral shedding and transmission by previously infected or vaccinated individuals [53] could prolong transition to stable endemic status. This transition could be impacted by many variables, including emergence of more transmissible or immune evasive variants [18, 53]. Viral transmission data derived from animal studies modeling natural exposure settings can provide a controlled experimental basis for SEIRS modeling during which the impact of variables such increased viral infectivity, immune avoidance and altered mixing ratios can be examined.

## Supporting information

**S1 Fig. Body weight by exposure mode after initial infection with SDAV.**
(DOCX)

**S2 Fig. Amount of viral shedding following initial SDAV exposure by exposure route.**
(DOCX)

**S3 Fig. Duration of viral shedding following initial SDAV exposure by exposure route.**
(DOCX)

**S4 Fig. Proportion of rats shedding SDAV per observation time by exposure route.**
(DOCX)

**S5 Fig. Relationship between amount of viral shedding and seroconversion.**
(DOCX)

**S6 Fig. Viral shedding on SDAV re-exposure.**
(DOCX)

**S1 Data.**
(XLSX)

## Acknowledgments

The authors thank members of the Yale Animal Resources Center for their excellent animal care.

## Author Contributions

**Conceptualization:** Caroline J. Zeiss.

**Data curation:** Brent Vander Wyk.

**Formal analysis:** Brent Vander Wyk, Heather G. Allore.

**Funding acquisition:** Caroline J. Zeiss.

**Investigation:** Caroline J. Zeiss, Jennifer L. Asher, Susan R. Compton.

**Methodology:** Caroline J. Zeiss, Jennifer L. Asher, Susan R. Compton.

**Project administration:** Caroline J. Zeiss.

**Resources:** Susan R. Compton.

**Supervision:** Caroline J. Zeiss.

**Visualization:** Caroline J. Zeiss.

**Writing – original draft:** Caroline J. Zeiss, Susan R. Compton.

**Writing – review & editing:** Jennifer L. Asher, Heather G. Allore, Susan R. Compton.

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
