## [Decision Letter · Decision Letter 0]

10 Sep 2021

PONE-D-21-19702Continued viral shedding and transmission after natural infection and vaccination in an animal model of SARS-CoV-2 propagationPLOS ONE

Dear Dr. Zeiss,

Thank you for submitting your manuscript to PLOS ONE. After careful consideration, we feel that it has merit but does not fully meet PLOS ONE’s publication criteria as it currently stands. Therefore, we invite you to submit a revised version of the manuscript that addresses the points raised during the review process.

Please revise your manuscript considering the issues raised by both expert reviewers regarding:TitleMethodologyPresentation of the resultsDescription of the coronavirus Sialodacryoadenitis Virus (SDAV) and the extension to SARS-CoV-2. The authors should be cautious and careful with human/SARS-CoV2 extrapolationDiscussionPlease ensure that your decision is justified on PLOS ONE’s publication criteria and not, for example, on novelty or perceived impact.

We look forward to receiving your revised manuscript.

Kind regards,

Graciela Andrei

Academic Editor

PLOS ONE

Journal Requirements:

Reviewers' comments:

Reviewer's Responses to Questions

**Comments to the Author**

1. Is the manuscript technically sound, and do the data support the conclusions?

Reviewer #1: Yes

Reviewer #2: Yes

2. Has the statistical analysis been performed appropriately and rigorously? 

Reviewer #1: Yes

Reviewer #2: Yes

3. Have the authors made all data underlying the findings in their manuscript fully available?

Reviewer #1: Yes

Reviewer #2: Yes

4. Is the manuscript presented in an intelligible fashion and written in standard English?

Reviewer #1: Yes

Reviewer #2: Yes

5. Review Comments to the Author

Reviewer #1: In this study, the authors tested the viral shedding and seroconversion following initial infection or reinfection with an infectious respiratory coronavirus, Sialodacryoadenitis Virus (SDAV) in rats. They also tested the replication and transmission of SDAV virus in rats vaccinated with a cross-protective coronavirus RVS. They found that portions of the seropositive previously infected or RVS-vaccinated animals can be infected with SDAV, shedding and transmitting virus to previously seronegative animals by direct contact. The phenomenon in infectious respiratory coronavirus SDAV in rats might occur in SARS-CoV-2 in humans as well. This study further our understanding of characterization of the infectious coronavirus.

Comments:

1. In this study, a rat infectious respiratory coronavirus SDAV, but not SARS-CoV-2, was used to infect rats. Situations in SDAV, which is a distant virus to SARS-CoV-2, can’t “model” those in SARS-CoV-2, even SARS-CoV or MERS-CoV virus can’t neither. The authors should focus on describing SDAV in Title and most parts of the main text. The extension to SARS-CoV-2 should be discussed carefully.

2. Line 81, “Rooms were maintained at 72oC” should be a typo.

3. As described in Table 1 and Line 200 “a negative PCR test does not rule out the potential of infection sufficient to induce an immune response”, the seronegative group generated in 1st animal experiment is mixed with infected and uninfected animals. These animals were used in 2nd animal experiments for contact exposure, considered as previously naïve animals. Please explain.

4. Last paragraph in Discussion, SARS-CoV-2 challenge experiment in vaccinated mink model (Shuai et al., National Science Revies, 2021) should be discussed.

Reviewer #2: PONE-D-21-19702: Continued viral shedding and transmission after natural infection and vaccination in an animal model of SARS-CoV-2 propagation

The manuscript of Zeiss et al., reports original and very interesting findings on viral shedding and viral transmission upon re-exposure in rats. The authors study the propagation of SDAV, the Sialodacryoadenitis Virus, a rat betacoronavirus affecting the upper airways, salivary and lacrimal glands and lungs in rats. Indeed, the authors tested different ways of viral exposures on rats via inoculation or direct contact with infected animals or with fomite contacts. They then re-exposed the rats to determine their viral shedding and transmission to naive animals. They finally compare their results with a population of rats that they have previously vaccinated with a related coronavirus (RCV), establishing there an interesting cross-virus immunity model. They find that both naturally infected and vaccinated animals can still propagate infection. Based on this animal model, the authors then extrapolate on immunity to SARS-CoV2 infection in humans.

The authors have chosen PLOS ONE as their publication target and place their objectives in an appropriate scientific context. Their experiments are well designed, and the results increase our current knowledge regarding viral shedding and transmission in an animal model, the rat.

My main concerns are : the title of the study could be misleading and the presentation of the methodology used and of the results obtained needs to be improved to facilitate the readers understanding of the different protocols used (infection-inoculation-vaccination). The authors could also easily generate 2 main figures out of Figures S1 and S2 to solve the issue.

Specific comments:

1) Even if the actual context of mass vaccination against COVID-19 is pressing, the authors could choose a title more representative of their study results which have been obtained in rats with rat viruses, such as, for example: “Modeling SARS-CoV-2 propagation using rat coronavirus-associated shedding and transmission”.

2) The introduction part of the manuscript is short. It mostly contains a summary of their study. Key references are missing.

3) The methods used (infection-inoculation-vaccination) should be clearly described and mentioned carefully throughout the manuscript to avoid confusion from the readers on the type of rats/viral exposure described. It would help to include the supplementary figures S1 and S2 (or a simplified version of S1 and S2) as main Figures.

4) To study the effects of “vaccination” on viral shedding and viral transmission, the authors established a “cross-virus immunity model”. Can we consider these animals as “vaccinated” as they have been exposed to RCV and not to SDAV?

5) Mat/Met question: The authors indicate numbers for the primers for RT-PCR experiments, such as SD29629. Are these numbers ordering numbers? From which company?

6) Mat/Met question: PCR cycles Cq is indicated as <40, this is high, any reason?

7) Mat/Met question: In the Tables, for the column entitled “PCR positive”, please indicate the n number of experiments as n/n experiments and not only the percentages.

8) Discussion: (p.16) The authors should be careful with human/SARS-CoV2 extrapolation and comment more on the interesting model they have established.

9) Discussion: (p.17) The use RT-PCR testing technique is not limited to SARS-CoV 2 surveillance.

10) Discussion: (p.19) Age plays an important role in viral susceptibility. Could the authors comment on working with rats with challenging protocols (vaccination protection) over weeks.

6. PLOS authors have the option to publish the peer review history of their article (what does this mean?). If published, this will include your full peer review and any attached files.

Reviewer #1: No

Reviewer #2: No

---

## [Author Response · Author response to Decision Letter 0]

21 Sep 2021

Dear Dr Andrei

We thank the reviewers for their insightful and very helpful comments. All have been addressed in full, and we feel the manuscript is significantly improved by their effort. Responses are detailed under each query below – in summary: 

1. We have refocused introduction and discussion significantly to clarify the extent to which we can draw comparisons between our model system and SARS-CoV-2 transmission in humans. Overall, we have taken a higher-level approach that discusses the utility of studying known coronaviruses to gain insight into newly emerged coronavirus, focusing on phylogenetic proximity, transmission dynamics and duration of immunity. 

2. We have edited the methods to more clearly describe our transmission strategies. 

3. We have revised Supp Figures 1 and 2 as Figures 1 and 2, screened these in PACE, and renumbered remaining Supp Figures accordingly. 

4. We have pointed out that our data can be used in SEIRS modeling, and have made all raw data freely available in Supp Material.

Reviewers' comments:

Reviewer #1: In this study, the authors tested the viral shedding and seroconversion following initial infection or reinfection with an infectious respiratory coronavirus, Sialodacryoadenitis Virus (SDAV) in rats. They also tested the replication and transmission of SDAV virus in rats vaccinated with a cross-protective coronavirus RVS. They found that portions of the seropositive previously infected or RVS-vaccinated animals can be infected with SDAV, shedding and transmitting virus to previously seronegative animals by direct contact. The phenomenon in infectious respiratory coronavirus SDAV in rats might occur in SARS-CoV-2 in humans as well. This study further our understanding of characterization of the infectious coronavirus.

Comments:

1. In this study, a rat infectious respiratory coronavirus SDAV, but not SARS-CoV-2, was used to infect rats. Situations in SDAV, which is a distant virus to SARS-CoV-2, can’t “model” those in SARS-CoV-2, even SARS-CoV or MERS-CoV virus can’t neither. The authors should focus on describing SDAV in Title and most parts of the main text. The extension to SARS-CoV-2 should be discussed carefully.

We have refocused the paper to clarify the extent to which we can draw comparisons between our model system and SARS-CoV-2 transmission in humans. Overall, we have taken a higher level approach that discusses the utility of studying known coronaviruses to gain insight into newly emerged coronavirus. Edits are made throughout, and are quite extensive in introduction and discussion.

We focus specifically on phylogenetic proximity, similar transmission dynamics and similar duration of immunity between SDAV and closely related human viruses HKU1 and OC43, and point out that patterns of HKU1 and OC43 transmission have been used to accurately predict SARS-CoV-2 transmission in humans. We are clear that SDAV is not an appropriate model of disease severity, but should only be used to model transmission, in the same way that HKU1 and OC43 are used to model SARS-CoV-2 transmission. Because coronaviruses are capable of fairly frequent cross-species transmission, transmission dynamics are often similar within and across species. 

We have removed some text in which specific comparisons are made between details of SDAV and SARS-CoV-2 pathogenesis eg incubation periods.

2. Line 81, “Rooms were maintained at 72oC” should be a typo.

Thank you - Corrected to 72oF

3. As described in Table 1 and Line 200 “a negative PCR test does not rule out the potential of infection sufficient to induce an immune response”, the seronegative group generated in 1st animal experiment is mixed with infected and uninfected animals. These animals were used in 2nd animal experiments for contact exposure, considered as previously naïve animals. Please explain.

This text refers to discordant results i.e. animals who are either PCR positive or seropositive, but not both. Animals defined as susceptible and used for inoculations in the reinfection experiment needed to be both seronegative and PCR negative. These animals were either mock inoculated (and thus established PCR and seronegative) or purchased (naïve). 

A total of 40 seronegative animals and 66 seropositive animals were used in the reinfection study – all seropositive animals from Table 1 were used (initially 67, but one died), but not all seronegative animals were used (initially 47 from Table 1 but 13 PCR negative seronegative animals were used in other experiments leaving 34). Of these, 18 animals (10 mock inoculated controls plus a balance of naïve purchased animals) were inoculated to provide enough reliably shedding animals to expose both seropositive and seronegative groups. 

13 animals from the initial experiment that were both seronegative and PCR negative (ie had never tested positive on oral swabs) were used as susceptible recipients in the third experiment (transmission study from SDAV reinfected rats) as we needed size/sex matched animals to direct contact house. We have more clearly defined susceptible (seroneg and PCR neg despite potential exposure in some cases) vs naïve (never exposed). 

The methods have been edited to reflect this more clearly. Additionally, we have included raw data from all experiments as Excel files in Supplementary Data. 

4. Last paragraph in Discussion, SARS-CoV-2 challenge experiment in vaccinated mink model (Shuai et al., National Science Revies, 2021) should be discussed.

We have addressed the issue of viral shedding (both detectable by PCR and isolation of live virus) and risk of transmission (particularly in cases of relatively low viral shedding) following coronaviral re-exposure in more detail as this is central to our paper. The mink paper is included, as are additional animal studies in hamsters and macaques 

Reviewer #2: PONE-D-21-19702: Continued viral shedding and transmission after natural infection and vaccination in an animal model of SARS-CoV-2 propagation

The manuscript of Zeiss et al., reports original and very interesting findings on viral shedding and viral transmission upon re-exposure in rats. The authors study the propagation of SDAV, the Sialodacryoadenitis Virus, a rat betacoronavirus affecting the upper airways, salivary and lacrimal glands and lungs in rats. Indeed, the authors tested different ways of viral exposures on rats via inoculation or direct contact with infected animals or with fomite contacts. They then re-exposed the rats to determine their viral shedding and transmission to naive animals. They finally compare their results with a population of rats that they have previously vaccinated with a related coronavirus (RCV), establishing there an interesting cross-virus immunity model. They find that both naturally infected and vaccinated animals can still propagate infection. Based on this animal model, the authors then extrapolate on immunity to SARS-CoV2 infection in humans.The authors have chosen PLOS ONE as their publication target and place their objectives in an appropriate scientific context. Their experiments are well designed, and the results increase our current knowledge regarding viral shedding and transmission in an animal model, the rat.

My main concerns are : the title of the study could be misleading and the presentation of the methodology used and of the results obtained needs to be improved to facilitate the readers understanding of the different protocols used (infection-inoculation-vaccination). The authors could also easily generate 2 main figures out of Figures S1 and S2 to solve the issue.

We have replaced the title with one suggested by this reviewer and revised the methods to more clearly explain our methodology and illustrate this with two main figures. 

Specific comments:

1) Even if the actual context of mass vaccination against COVID-19 is pressing, the authors could choose a title more representative of their study results which have been obtained in rats with rat viruses, such as, for example: “Modeling SARS-CoV-2 propagation using rat coronavirus-associated shedding and transmission”.

Thank you – we like this title and have used it 

2) The introduction part of the manuscript is short. It mostly contains a summary of their study. Key references are missing.

The introduction has been expanded to include more information on transmission, disease phenotype and extent of immunity to SDAV. We also clarify the extent to which we can draw comparisons between out model system and SARS-CoV-2 transmission in humans. We focus specifically on phylogenetic proximity, similar transmission dynamics and similar duration of immunity between SDAV and closely related human viruses HKU1 and OC43, and point out that patterns of HKU1 and OC43 transmission have been used to accurately predict SARS-CoV-2 transmission in humans. We are clear that SDAV is not an appropriate model of disease severity, but should only be used to model transmission, in the same way that HKU1 and OC43 are used to model SARS-CoV-2 transmission. Because coronaviruses are capable of fairly frequent cross-species transmission, transmission dynamics are often similar within and across species. More references have been included. 

3) The methods used (infection-inoculation-vaccination) should be clearly described and mentioned carefully throughout the manuscript to avoid confusion from the readers on the type of rats/viral exposure described. It would help to include the supplementary figures S1 and S2 (or a simplified version of S1 and S2) as main Figures.

Figures S1 and S2 have been used to create Figures 1 and 2 (they have not been combined as they are fairly complex however Fig 2 has been simplified). Figure legends and text have been rewritten for clarity - the used of ”index” has been replaced with “inoculated” throughout.

4) To study the effects of “vaccination” on viral shedding and viral transmission, the authors established a “cross-virus immunity model”. Can we consider these animals as “vaccinated” as they have been exposed to RCV and not to SDAV?

We based our heterologous vaccination approach on prior studies by Bihun CG, Percy DH. Coronavirus infections in the laboratory rat: degree of cross protection following immunization with a heterologous strain. Can J Vet Res. 1994 Jul;58(3):224-9. 

In this study, rats were immunized with parkers rat coronavirus, a rat virus that is antigenically closely related to SDAV (to the extent that seroconversion to either RCV or SDAV can be assessed using the same IFA assay) and which causes a generally similar spectrum of illness. Vaccinated rats were challenged with SDAV 3 and 6 months later (at which point all vaccinated animals were still seropositive). Vaccinated animals were significantly but not entirely protected against SDAV challenge, and exhibited mild histologic lesions but no clinical illness. 

This is explained more fully in the methods “Modeling Vaccination using RCV”

5) Mat/Met question: The authors indicate numbers for the primers for RT-PCR experiments, such as SD29629. Are these numbers ordering numbers? From which company?

The primers were designed in house and were purchased from the Keck oligonucleotide synthesis facility at Yale. The numbers correspond to the nucleotide positions in the SDAV genome (JF92616). This has been added in the text 

6) Mat/Met question: PCR cycles Cq is indicated as <40, this is high, any reason?

The RT-PCR program used includes 40 amplification cycles, and is a standard procedure in our comparative virology diagnostic lab. 

7) Mat/Met question: In the Tables, for the column entitled “PCR positive”, please indicate the n number of experiments as n/n experiments and not only the percentages.

Tables 1-3 have been updated as suggested. 

8) Discussion: (p.16) The authors should be careful with human/SARS-CoV2 extrapolation and comment more on the interesting model they have established.

As described above, we have taken this advice to heart. We have refocused the paper to clarify the extent to which we can draw comparisons between our model system and SARS-CoV-2 transmission in humans. Overall, we have taken a higher level approach that discusses the utility of studying known coronaviruses to gain insight into newly emerged coronavirus. We have eliminated some direct comparisons between SARS-CoV-2 and SDAV. Edits are made throughout, and are quite extensive in introduction and discussion.

9) Discussion: (p.17) The use RT-PCR testing technique is not limited to SARS-CoV 2 surveillance.

We have clarified this as follows: “We used the two most widely used measures of SARS-CoV-2 (and other viral) surveillance,”

10) Discussion: (p.19) Age plays an important role in viral susceptibility. Could the authors comment on working with rats with challenging protocols (vaccination protection) over weeks.

Our goal was to model asymptomatic transmission (not disease, in which there is certainly a very clear age susceptibility for COVID-19) in young adults, an important source of SARS CoV-2 propagation. Published work on SDAV describes infections in animals from 8 weeks to up to 15 months of age. No sex, strain or age-based differences in susceptibility have been described for SDAV and RCV within this age span. In our study, animals were initially infected at 8 weeks of age, and reinfected after 165 days or less. Re-exposed (older) susceptible animals demonstrated very similar infection rates to those seen in initially infected (younger) naïve animals. Therefore, it is unlikely that the months between initial and subsequent infection significantly influenced susceptibility to infection due to age. SDAV demonstrates broad infectivity when introduced into naïve populations, resulting in epidemic morbidity followed by sporadic outbreaks in the enzootic state. Therefore our data can be used to model transmission events in young adults, influenced primarily by prior immune exposure. 

To ensure that this is clear to the reader we have included ages at euthanasia to the end of each methods section. 

References:

 15 months: Percy DH, Bond SJ, Paturzo FX, Bhatt PN. Duration of protection from reinfection following exposure to sialodacryoadenitis virus in Wistar rats. Lab Anim Sci. 1990 Mar;40(2):144-9.

8 weeks: Percy DH, Scott RA. Coronavirus infection in the laboratory rat: immunization trials using attenuated virus replicated in L-2 cells. Can J Vet Res. 1991 Jan;55(1):60-6.

11 months: Weir EC, Jacoby RO, Paturzo FX, Johnson EA. Infection of SDAV-immune rats with SDAV and rat coronavirus. Lab Anim Sci. 1990 Jul;40(4):363-6.

9 weeks: Percy DH, Hanna PE, Paturzo F, Bhatt PN. Comparison of strain susceptibility to experimental sialodacryoadenitis in rats. Lab Anim Sci. 1984 Jun;34(3):255-60.

---

## [Decision Letter · Decision Letter 1]

2 Nov 2021

Modeling SARS-CoV-2 propagation using rat coronavirus-associated shedding and transmission

PONE-D-21-19702R1

Dear Dr. Zeiss,

We’re pleased to inform you that your manuscript has been judged scientifically suitable for publication and will be formally accepted for publication once it meets all outstanding technical requirements.

Kind regards,

Graciela Andrei

Academic Editor

PLOS ONE

Additional Editor Comments (optional):

Reviewers' comments:

Reviewer's Responses to Questions

**Comments to the Author**

1. If the authors have adequately addressed your comments raised in a previous round of review and you feel that this manuscript is now acceptable for publication, you may indicate that here to bypass the “Comments to the Author” section, enter your conflict of interest statement in the “Confidential to Editor” section, and submit your "Accept" recommendation.

Reviewer #1: All comments have been addressed

Reviewer #2: All comments have been addressed

2. Is the manuscript technically sound, and do the data support the conclusions?

Reviewer #1: Yes

Reviewer #2: Yes

3. Has the statistical analysis been performed appropriately and rigorously? 

Reviewer #1: Yes

Reviewer #2: Yes

4. Have the authors made all data underlying the findings in their manuscript fully available?

Reviewer #1: Yes

Reviewer #2: Yes

5. Is the manuscript presented in an intelligible fashion and written in standard English?

Reviewer #1: Yes

Reviewer #2: Yes

6. Review Comments to the Author

Reviewer #1: This study further our understanding of characterization of the infectious coronavirus. All comments have been addressed and the manuscript is significantly improved.

Reviewer #2: PONE-D-21-19702R1

The authors have done a significant rewriting of their manuscript. They appropriately rephrased and reoriented the description and purpose of their paper. They responded in a relevant way to all the questions and comments raised by the reviewers. The addition in the text of the explanations and the comparisons with the HKU1 and OC43 viruses is particularly appropriate and thus makes the reading both fluid and oriented. The authors have also made a great effort by emphasizing the limits of their approach and comparisons with SARS-CoV-2 as well as by including the latest epidemiological notions of COVID-19, such as the variants.

After reading this new manuscript, a number of questions / remarks still emerge. I leave it to the authors / editors to take them into account (or not) in the final stages of writing.

I highly recommend this manuscript for publication as a Research article in Plos One and congratulate the authors for their work.

Final questions / remarks:

1) Is the phenomenon of heterologous vaccination also observed in humans for HKU1 / OC43 viruses for the SARS-CoV-2? This point could be raised in the discussion.

2) In my opinion, a reference concerning the 40 amplification cycles used as a “standard” would further legitimize their approach.

3) For consistency, the authors should use the same temperature units and symbols (e.g. "°C") throughout the text (e.g. line 105 vs. lines 228-229).

7. PLOS authors have the option to publish the peer review history of their article (what does this mean?). If published, this will include your full peer review and any attached files.

Reviewer #1: **Yes: **Hualan Chen

Reviewer #2: No

---

## [Editor Report · Acceptance letter]

15 Nov 2021

PONE-D-21-19702R1 

Modeling SARS-CoV-2 propagation using rat coronavirus-associated shedding and transmission 

Dear Dr. Zeiss:

I'm pleased to inform you that your manuscript has been deemed suitable for publication in PLOS ONE. Congratulations! Your manuscript is now with our production department. 

Kind regards, 

on behalf of

Dr. Graciela Andrei 

Academic Editor

PLOS ONE